# Skin-Aging Pigmentation: Who Is the Real Enemy?

**DOI:** 10.3390/cells11162541

**Published:** 2022-08-16

**Authors:** Jin Cheol Kim, Tae Jun Park, Hee Young Kang

**Affiliations:** 1Department of Dermatology, Ajou University School of Medicine, Suwon 16499, Korea; 2Department of Biochemistry and Molecular Biology, Ajou University School of Medicine, Suwon 16499, Korea; 3Inflamm-Aging Translational Research Center, Ajou University School of Medicine, Suwon 16499, Korea; 4Department of Medical Sciences, Ajou University Graduate School of Medicine, Suwon 16499, Korea

**Keywords:** aging, cellular senescence, fibroblasts, keratinocytes, melanocytes, senescence, skin pigmentation

## Abstract

Skin aging is induced and sustained by chronological aging and photoaging. Aging skin pigmentation such as mottled pigmentation (senile lentigo) and melasma are typical signs of photoaging. The skin, like other human organs, undergoes cellular senescence, and senescent cells in the skin increase with age. The crosstalk between melanocytes as pigmentary cells and other adjacent types of aged skin cells such as senescent fibroblasts play a role in skin-aging pigmentation. In this review, we provide an overview of cellular senescence during the skin-aging process. The discussion also includes cellular senescence related to skin-aging pigmentation and the therapeutic potential of regulating the senescence process.

## 1. Introduction

Aging is a heterogeneous and time-dependent process that is influenced by various factors, including genetic and environmental components, gradually progressing in keeping with chronological age in humans [1]. The aging process occurs in multiple organs at different rates and degrees, and aging organs show an accumulation of senescent cells. Skin aging is clinically characterized by pigmentation, atrophy, a loss of elasticity, and an impaired recovery response against damage, resulting in subsequent pathologic skin disorders. The aging of human skin can be induced by intrinsic and extrinsic pathways. Genetic and hormonal factors are intrinsically associated with unavoidable skin aging, while environmental factors such as ultraviolet radiation are extrinsically associated with preventable skin aging. Recently, skin pigmentation during the aging process has been actively discussed, and cellular senescence (i.e., a state of permanent cell-cycle arrest) is thought to be a key player in skin-aging pigmentation. Various types of skin cells, such as keratinocytes, melanocytes, fibroblasts, and endothelial cells, are involved in skin aging, and the crosstalk between these cells during the aging process may play an important role in melanogenesis and the subsequent aging-related pigmentation. In other words, various factors and cells influence each other; thus, it is difficult to designate a specific cause or the main culprit causing skin-aging pigmentation. In this review, we aim to provide an overview of age-dependent changes in senescent skin cells and skin pigmentation and to discuss how cellular senescence influences the occurrence of skin-aging pigmentation. Additionally, we discuss whether the regulation or interruption of cellular senescence in human skin can prevent the skin from aging or delay the process, and we provide promising therapeutic targets such as senolytics and senomorphics for skin-aging pigmentation.

## 2. Senescent Cell Markers and Age-Dependent Changes in Skin Cells

The number of senescent cells increases with chronological age, and the age-related increase in the number of senescent cells was found to be associated with the pathogenesis of various diseases [2,3]. In the skin, fibroblasts and melanocytes are known to be the most observed senescent skin cells, and the numbers of these cells increase in an age-dependent manner [4,5,6]. These phenomena related to senescent cells have been identified using senescent cell markers at the cellular level. An exploration of morphological changes in cells can be helpful to detect senescent cells, which are mainly characterized by an enlarged, flattened, and irregular shape [7,8]. The mTOR signal pathway is associated with the enlargement of the cell body and an increased level of protein synthesis in relation to mTORC1 activation [9,10]. In skin cells, senescent keratinocytes [11] and UVB-induced senescent melanocytes [12], as confirmed by other senescent cell markers, show an increase in the cells’ size. Moreover, an increase in the granular content in senescent fibroblasts and melanocytes has been observed as a result of the intracellular deposition of lipofuscin in lysosomes and glycogen particles [13,14,15,16]. In addition, cytoskeleton rearrangement, especially in relation to vimentin filaments, can influence the cells’ shape during senescence [13,17,18], while in the senescent state, cells show the upregulation of lysosomal contents and enzymes [19]. Senescence-associated β-galactosidase (SA-β-Gal), a lysosomal enzyme, is most commonly increased in senescent cells and is one of the most widely used markers for analyzing senescent cells in cultures and freshly frozen tissue samples [4,20,21,22,23], despite its limited applicability with frozen tissues or live cells. However, because not all senescent cells always express higher activity levels of SA-β-Gal [24], the combined use of other senescence markers may be necessary.

Cyclin-dependent kinases (CDKs) play important roles in the regulation of cell division and modulation of cell transcription [25]. CDK inhibitors, including the INK4 or CIP/KIP protein families, are associated with the cell-cycle arrest. INK4 proteins such as p16^INK4a^, p15^INK4b^, and p18^INK4c^ distort the cyclin interface and the ATP-binding pocket and subsequently prevent CDK4/CDK6 activation through D-type cyclins [26]. Furthermore, CIP/KIP proteins such as p21^CIP1^, p27^KIP1^, and p57^KIP2^ inhibit CDK/cyclin heterodimers [27]. Among them, p16^INK4a^, p15^INK4b^, and p21^CIP1^ are known to be key players in the cell-cycle arrest process during cellular senescence. In multiple organ tissues, p16^INK4a^-positive senescent cells increase, particularly in cases of aging-related diseases [2,28,29]. Especially in the skin of elderly humans, the number of p16^INK4a^-positive cells is increased in the epidermis and dermis compared with that in younger age groups [29]. The induction of p16^INK4a^ is often triggered by epigenetic changes [30,31], promotor accessibility [32,33], and protein stability [34]. Although p21^CIP1^ is also upregulated in aging cells [35], it can be activated directly through the p53 effect and activated regardless of p53 as well, through several transcription factors [36]. Therefore, unlike p16^INK4a^, using p21^CIP1^ as a single marker of cellular senescence may be problematic. Moreover, p21^CIP1^ is downregulated in senescent melanocytes [37]. These factors may contribute to the more popular use of p16^INK4a^ for detecting senescent cells in various experimental studies. Very recently, an in vivo study by Kim et al. (5) revealed age-dependent changes in senescent cells in the skin. Their findings showed that fibroblasts in sun-exposed areas of the skin more frequently exhibit p16^INK4a^ immunoreactivity compared with those in the unexposed skin. Especially in the exposed skin, the senescence of fibroblasts is initiated at a younger age, though it has been observed in all age groups, while the senescence of melanocytes is observed only from middle age, and a few p16^INK4a^ positive keratinocytes are observed in the elderly skin. Although the senescence of fibroblasts precedes that of melanocytes, all senescent skin cells show an age-related sequential increase in the exposed skin. Furthermore, the parallel senescence of melanocytes and their adjacent cells, such as keratinocytes and fibroblasts, has been observed. Taken together, the crosstalk between senescent skin cells may play a critical role in relation to changes in age-associated phenotypes in aging skin. Exploring and understanding the cellular and neighboring microenvironmental changes in skin cells during the aging process are, thus, important endeavors.

When a cell enters a senescent state, it releases various pro-inflammatory cytokines, chemokines, growth factors, and proteinases such as matrix metalloproteinases (MMPs), referred to as the senescence-associated secretory phenotypes (SASPs). This procedure induces the target cells to be senescent and changes the microenvironment of their regional tissue [38,39]. Although the compositions and quantities of SASPs can differ according to the types of cells and the initial stimuli, the main components of SASPs in senescent fibroblasts and melanocytes consist of interleukin (IL)-6, IL-8, CXC chemokine ligand (CXCL) 2, MMP3, MMP9, and insulin-like growth-factor-binding protein 7 (IGFBP7) [39,40]. These SASP factors are mainly induced by the transcription factor NF-κB with the DNA damage response (DDR), but other factors such as the cGAS/STING pathway [41,42,43], GATA-binding protein 4 (GATA4) [44], and CCAAT-/enhancer-binding protein beta (C/EBPβ) [45,46,47] can also lead to SASP-medicated cellular senescence.

## 3. Triggering Factors Driving Cellular Aging

Skin aging involves many complex factors and processes, as with other organs. It is widely accepted that the causal factors of skin aging can be divided simply into two types: intrinsic and extrinsic. Intrinsic skin aging, also called chronologic skin aging, refers to the physiologic changes in the skin occurring with time, clinically characterized as dry, pale skin with fine wrinkles and less elasticity. This type is also modulated by hormonal factors such as sex steroids [48,49]. Decreased estrogen levels and their receptors in epidermal keratinocytes and dermal fibroblasts over time cause and aggravate the skin changes mentioned above. In terms of skin pigmentation, an acquired chronic relapsing hyperpigmented disease such as melasma can be improved as the estrogen level decreases, because the estrogen underlying UVB exposure can maintain pigmentation by increasing the number of blood vessels [50]. While intrinsic aging occurs at some point in most human organs, including the skin, extrinsic skin aging mostly occurs in environmentally exposed skin areas such as the face, neck, or dorsum of the hands. External environmental factors, collectively defined as the “exposome”, such as UVB and UVA radiation, infrared and visible light, air pollution, cigarette smoking, chemical exposure, diet, stress, sleep loss, and trauma, can cause extrinsic skin aging [51]. As extrinsic skin aging is mostly mediated by UV radiation (i.e., “photoaging”), which can lead to reactive oxygen species (ROS) and DNA damage in the cells, clinically, it is associated with coarse wrinkles, and irregular pigmentation, and lentigines [52]. Therefore, it can be clinically differentiated from intrinsic skin aging.

Specifically, aging at the cellular level may be triggered by telomere shortening, considered to be replicative senescence, and by responses to various stimuli such as DNA damage, oxidative stress, oncogenes, mitochondrial dysfunction, epigenetic changes, and paracrine factors. The term replicative senescence has been used to describe the reduction in the proliferative capacity after multiple cell division, subsequently resulting in the completely irreversible arrest of cell growth [53]. This type of cellular senescence was initially discovered through the limited in vitro proliferation of human fibroblasts [7,54]. During replicative senescence, the shortening of telomeres is an important causal factor [55]. When telomeres are critically shortened, the persistent activation of DDR pathways is triggered, resulting in replicative cellular senescence [56].

Sufficient DNA damage with the cell not proceeding to the stage of apoptosis can induce cell-cycle arrest in live cells [57,58]. Various factors such as ultraviolet (UV) or ionizing radiation can cause DNA-damage-induced cellular senescence [58], and the mechanisms of this type of cellular senescence mostly proceed during the process of double-strand DNA breaks (DSBs), followed by DDR. DDR, as an underlying cause of cellular senescence, is mainly induced by DNA breaks, which promote DDR signaling kinases such as ataxia-telangiectasia-mutated (ATM) kinase and ATR through the phosphorylation of histone H2AX [59,60]. In response to these kinase-medicated DNA-damage signals, the subsequent activation of p53 induces other cyclin-dependent-kinase (CDK) inhibitors such as p21 and p16 in the early and late stages, respectively [61,62].

Cellular senescence secondary to oxidative damage is also involved in aging [63]. Free radical theory in relation to the aging process was initially introduced in 1954 by Harman et al. [64]. Oxidative stress can lead to the upregulation of stress-induced factors and subsequently various cytokines, including hypoxia-inducible factors (HIFs), nuclear factor κB (NF-κB), interleukin (IL)-1, IL-6, vascular endothelial growth factor (VEGF), and tumor necrosis factor (TNF)-α. These factors act as proinflammatory agents in senescent cells [65,66,67]. Additionally, oxidative damage to DNA can modify the normal structure of telomeres through the disruption of the 3′ overhang at the end of the telomeres, prematurely resulting in p53-signaling-mediated proliferative senescence or apoptosis [68], meaning that DSBs within not critically shortened telomeres (i.e., telomeric DNA damage) can induce cellular senescence regardless of the telomere’s length [69,70,71].

Oncogenic activation, such as RAS or BRAF, engaging the DDR pathways is another cause of cellular senescence [72,73], known as oncogene-induced senescence (OIS). The loss of PTEN as a tumor suppressor can induce hyperproliferation with DDR, resulting in cellular senescence [74]. Unlike OIS, with DDR pathways, p53-dependent cellular senescence is mediated by the activation of the PI3K–AKT pathway without DDR [75,76]. Furthermore, mitochondrial dysfunction is a trigger factor for cellular senescence (i.e., mitochondrial-dysfunction-associated senescence (MiDAS)) through the NADH–AMPK–p53-dependent pathway. Additionally, chromatin structure changes [77,78], and epigenetic stress [79] can induce cellular senescence.

## 4. Role of Cellular Senescence in Skin-Aging Pigmentation

With chronological aging, the number of functional melanocytes gradually declines, and melanogenic activity, including tyrosinase activity, is reduced, resulting in the appearance of pale skin in the elderly [80]. However, the melanocytes of photoexposed skin are relatively well-maintained compared with those of unexposed aging skin, most likely due to persistent UV stimuli affecting melanocytes [81,82]. Although it may be considered a protective mechanism to prevent the skin from photodamage, the resulting morphological and functional phenotype changes in melanocytes are induced, and subsequent, pigmentary disorders develop. The representative pigmentary changes during the aging process include senile lentigo and melasma. Senile lentigo is often observed in the elderly and most commonly occurs on exposed skin areas such as the face and dorsum of the forearm or hand. It is characterized by variably sized light- to dark-brownish macules and patches. Melasma is another representative pigmentary disorder that develops on sun-exposed skin areas, although it is not an age-related disease [83]. Melasma shows clinically diffuse light- to dark-brown pigmentation on centrofacial and malar regions as a chronic and relapsing condition [84]. Mottled pigmentation includes idiopathic guttate hypomelanosis (IGH) characterized by multiple hypopigmented well-circumscribed tiny macules usually localized on the chronically photoexposed areas of the forearms and legs. It is known to develop more frequently with age [85].

In the hyperpigmented skin of senile lentigo and melasma, active melanocytes are the major culprits of increased pigmentation. The hypopigmentation of IGH is due to the decreased number of melanocytes and to reduced melanin contents [86]. However, the causes of skin-aging pigmentation are not only restricted to melanocytes themselves. Typical neighboring skin cells, including keratinocytes, fibroblasts, and endothelial cells, also interact with pigmentary cells during the disruption of the pigmentation system’s homeostasis. Therefore, understanding cellular crosstalk between melanocytes and the neighboring cells in skin-aging pigmentation is necessary to investigate the role of cellular senescence in various pigmentary diseases. However, it should also be considered that a single transduction process or signaling pathway can be crucial for developing skin-aging pigmentation in certain congenital or acquired hyperpigmented and hypopigmented diseases [87].

### 4.1. UV-Irradiated Keratinocytes Activate Melanocytes, Resulting in a Tanning Response

The roles of epidermal keratinocytes in skin pigmentation have been well-described with an inducible tanning response after UV exposure, which has two components according to the applied wavelengths. The first is immediate tanning mainly in response to UVA radiation, and the other is delayed tanning mainly in response to UVB and shorter-wavelength UVA radiation [88,89]. The exposure of keratinocytes to UV radiation, especially UVB, induces DNA damage and subsequent p53 activation, resulting in the p53-mediated upregulation of proopiomelanocortin (POMC). This leads to the increased production of α-melanocyte-stimulating hormone (α-MSH) as a post-translational cleavage product of POMC in keratinocytes, after which α-MSH stimulates the melanocortin 1 receptors (MC1Rs) on adjacent melanocytes, resulting in the activation of microphthalmia-associated transcription factor (MITF) and melanogenesis [88,90,91,92]. Additionally, the secretion of endothelin-1 (ET-1), basic fibroblast growth factor (bFGF), and stem cell factor (SCF) from keratinocytes activates the endothelin B receptor (ENDRB), bFGF receptor tyrosinase kinase (RTK), and c-kit, respectively, resulting in the stimulation of melanogenesis [93]. These signal transductions are reportedly triggered by proinflammatory cytokines released from UV-exposed keratinocytes, such as IL-1α and tumor necrosis factor (TNF)-α [94]. In addition, the UV-radiation-mediated downregulation of transforming growth factor (TGF)-β1 [95] and the upregulation of adenosine 5′-triphosphate (ATP) production [96] in keratinocytes also result in increased MITF and melanogenesis through the TGF-β–PAX3 and ATP–P2X7 signaling pathways, respectively.

Indeed, p16^INK4a^-positive keratinocytes in normal skin are very rarely observed, highlighting the nature of the continuous proliferation and differentiation of the keratinocytes in the skin [5]. However, senescent changes in keratinocytes have been found in senile lentigo skin. Senile lentigo skin is characterized by a thickened epidermis, possibly due to the enlargement of individual keratinocytes and not to changes in their numbers [97,98]. Anti-p16 antibody staining is more intense in the lesional epidermis, suggesting the senescence of keratinocytes in the thickened epidermis [97]. It has been suggested that senile lentigo has a shared genetic basis with seborrheic keratosis, in which the enhanced expression of p16 indicates that keratinocytes are in a senescent state [99,100]. However, the roles of senescent keratinocytes and their SASPs in aging-associated pigmentary changes are still questionable. SASPs such as TNF-α, IL-1α, IL-1β, and IL-6 are also secreted from UV-induced senescent keratinocytes in relation to the increased level of NF-κB and a decrease in Y-box-binding protein 1 [101,102]. Additionally, autophagy impairment can result in skin-aging pigmentation. Reduced heat shock 70 kDa protein 1A (Hsp70-1A) and accelerated keratinocyte senescence impair autophagy and induce intracellular protein aggregation by reducing melanosome degradation in hyperpigmentation [87,103,104], and melatonin-induced autophagy activation protects keratinocytes from oxidative stress through the silent information regulator 1 (SIRT1) pathway [105]. However, defect in autophagy due to ectopic P-granules autophagy protein 5 homolog (EPG5) or tuberous sclerosis complex (TSC) mutations can lead to hypopigmentation [87].

### 4.2. Senescent Fibroblasts Contribute to the Development of the Hyperpigmentation of Senile Lentigo and Melasma

Various factors, including UV, can trigger the senescence of fibroblasts, and SASPs developed from senescent fibroblasts, such as IL-1, IL-6, and IL-8, as well as MMPs, can aggravate the aging process [106]. Fibroblast senescence is initiated in young skin and increases significantly with age [5]. The roles of senescent fibroblasts in skin-aging pigmentation have been well-described. Interestingly, hyperpigmented skin cases such as melasma and solar lentigo show a greater accumulation of senescent fibroblasts in dermal skin compared with perilesional normal skin [107,108]. Fibroblasts from photoaged skin express high levels of pro-melanogenic growth factors, such as hepatocyte growth factor (HGF), keratinocyte growth factor (KGF), and SCF [98,109,110,111,112]. In addition, several melanogenesis-modulating factors derived by senescent fibroblasts have been discovered in relation to skin-aging pigmentation. The hyperpigmented skin of melasma patients expresses high levels of Wnt inhibitory factor-1 (WIF-1) compared with perilesional normal skin [113]. Secreted frizzled-related protein 2 (sFRP2), functioning as a melanogenic stimulator through β-catenin signaling in human melanocytes, is overexpressed in melasma, solar lentigo, and acutely UV-irradiated skin, and fibroblast-derived sFRP2 may be closely related to the regulation of skin pigmentation [114]. Through epigenetic changes, senescent fibroblasts with decreased stromal-derived factor 1 (SDF1) expression contribute to aging pigmentation [108]. The growth differentiation factor 15 (GDF15) expression is increased in senescent fibroblasts, and photoaged, hyperpigmented skin is induced by senescent fibroblast-derived GDF15 through β-catenin signaling [115]. Taken together, these findings suggest that senescent fibroblasts are major players with regard to the continuous activation of melanocytes, which may result in the increased pigmentation of photoaged, pigmented skin.

### 4.3. Senescent Changes in Melanocytes in the Aging Skin

The regulatory role of senescent melanocytes in skin-aging pigmentation is very limited. Rather, the senescent melanocytes appear to be a victim of the increased melanogenesis that occurs during chronic sun exposure. During UV irradiation, the neighboring cells such as keratinocytes and fibroblasts release variable melanogenic factors, resulting in increased melanin production. Although it is minor, UV irradiation directly activates melanocytes, which leads to increased melanin pigment through prolonged p53 expression [16]. Repetitive melanocyte activation in photoexposed skin can result in senescent changes in melanocytes. Melanin accumulation repeatedly accelerates melanocyte senescence [116]. Indeed, senescent melanocytes appear on the skin of people, generally in their 40s [5]. The melanocytes from IGH patients are characterized by retracted or less-developed dendrites and reduced melanin transfer to adjacent keratinocytes, resulting in the cytoplasmic accumulation of melanin [86]. This reflects the role of senescence of melanocytes in hypopigmented aging skin.

In a recent study, apart from their role in the development of aging pigmentation, senescent melanocytes were suggested to have a propagator role in skin aging via paracrine telomere dysfunctions of keratinocytes and fibroblasts. Senescent melanocytes alone can express various inflammatory markers and SASPs, and these factors can trigger telomere dysfunctions. This process subsequently results in the proliferation arrest of adjacent cells such as keratinocytes through CXCL3-dependent mitochondrial ROS and induces skin aging [117]. With regard to the roles of senescent melanocytes in skin-aging pigmentation, further studies are necessary. Nonetheless, senescent melanocytes may contribute to skin-aging hypopigmentation and the senescence of neighboring skin cells.

### 4.4. Other Cells

Endothelial cells may also be involved in the regulation of skin-aging pigmentation. A previous study revealed that the number and size of dermal blood vessels both increase and that VEGF expression is upregulated in melasma and senile lentigo, compared with perilesional normal skin [118]. Considering that UV exposure induces angiogenesis with the expression of VEGF, bFGF, and IL-8 [119], photoaged skin disorders may develop due to vascular alterations. Previous in vitro co-culture models using endothelial cells and melanocytes also showed that ET-1 released by microvascular endothelial cells activates the ET-1/ENDRB signaling pathway, resulting in increased melanogenesis [120]. Additionally, other work showed that UVB-radiation-stimulated human dermal endothelial cell expresses inducible nitric oxide synthase (iNOS) as a melanogenic factor [121]. In addition, UV-irradiated dermal endothelial cells induce skin pigmentation by the paracrine stimulation of melanocytes through the upregulation of SCF [122].

The senescence of immune cells may be involved in skin aging. Photoaged skin shows higher levels of components of mononuclear phagocytes (MNPs), such as macrophages and monocytes, and the C–C motif chemokine ligand 2 (CCL2) derived from senescent fibroblasts also leads to the increased levels of prostaglandin E2-producing monocytes, resulting in the chronic inflammation of the skin. Additionally, senescent T cells release various cytokines [123]. However, the relationship between immunosenescence and skin-aging pigmentation has not been thoroughly investigated.

## 5. Therapeutic Strategy for Skin-Aging Pigmentation

Recently, various therapeutic approaches that interfere with the skin-aging process have been actively discussed [124]. Indeed, widely used clinical approaches to manage skin aging and skin-aging-related disorders in the real world have focused on improvements in undesirable pathologic consequences, such as melanin pigment accumulation, dermal collagen degradation, or loss of elasticity [125]. However, these whitening and dermal remodeling or rejuvenation modalities are not the ultimate treatment methods for preventing or reversing skin aging, and it has been suggested that skin-aging-related disorders can relapse unless senescent cells—the real enemy in skin-aging phenotypes—are removed. Therefore, targeting senescent cells and the related factors may be a more important treatment strategy to prevent and reverse skin-aging phenotypes (Figure 1). Strategies for inhibiting senescent-cell-mediated responses can be divided into two categories: senolytics and senomorphics. The selective induction of cell death specifically to kill senescent cells using drugs is referred to as senolytics, whereas senomorphics can regulate the microenvironment or extrinsic factors without the elimination of senescent cells [79].

To prevent skin-aging pigmentation, several possible senotherapeutics in senescent cells, especially senescent fibroblasts, have been suggested in previous studies [108,114,115,126], as have several possible candidate factors such as target-regulating SASPs. Based on the evidence that senescent fibroblasts can induce skin-aging pigmentation through the upregulation of SFRP2 [114] and GDF15 [115] and the downregulation of SDF1 [108] and clusterin (CLU) [126], the control of these SASPs derived by senescent fibroblast may be an effective therapeutic method to improve and prevent aging and aging-related skin pigmentation. The inhibition of SFRP2 and GDF15 can suppress melanogenic stimulation through β-catenin signaling in normal human melanocytes [114,115]. SDF1 can act as an anti-melanogenic agent by the downregulation of cAMP/phosphor-CREB/MITF/tyrosinase signaling in melanocytes [108]. Additionally, when comparing the concentrated conditioned medium from fibroblasts infected with a CLU-lentivirus with control, the melanin contents and tyrosinase activity were significantly reduced [126]. Another senomorphics agent, quercetin as a proteasome activator, has a restoring effect on the senescence of human dermal fibroblasts by reducing intracellular or extracellular ROS levels and increasing the gene expression levels of antioxidant enzymes [127], also having a whitening effect [128]. Therefore, these compounds can also be considered during the treatment of skin-aging pigmentation.

The elimination of senescent fibroblasts has been found in vivo using a radiofrequency treatment targeting the dermis to reduce pigmentation [108]. In these studies, a nonspecific reduction in p16^INK4A^-positive senescent fibroblasts and the stimulation of collagen production after radiofrequency contribute to a decrease in epidermal pigmentation. Therefore, the elimination of senescent fibroblasts has the potential to improve photoaging-related skin pigmentation and re-lighten pigmented skin. The beneficial effects of senolytics in improving skin aging were recently demonstrated [129,130]. Topical rapamycin, an FDA-approved drug targeting the mTOR complex, reduces either the expression of the p16^INK4A^ protein consistent with a reduction in cellular senescence or the number of cells entering senescence with clinical improvements in aging skin [131]. In addition, treatments with ABT-263 or ABT-737, well-known B-cell lymphoma 2 (BCL-2) inhibitors and senolytics, induced the selective clearance of senescent dermal fibroblasts [129]. The aged mouse skin treated with ABT-263 or ABT-737 showed increased collagen density, epidermal thickness, and the proliferation of keratinocytes, as well as decreased levels of senescence-associated secretory phenotypes, such as MMP-1 and IL-6. Furthermore, Park et al. [130] recently revealed that ABT-263 has anti-melanogenic effects and skin-lightening capabilities through their selective senolytic activity on senescent fibroblasts. ABT-737 also induced the clearance of senescent melanocytes in in vivo 3D human epidermal equivalents, subsequently preventing the neighboring cells from becoming senescent [132]. However, these BCL-2 inhibitors can potentially have harmful effects on skeletal-lineage cells such as trabecular bone loss [133]. Therefore, further investigative research to determine proper drug doses and delivery methods in relation to the safety and reliability of senotherapeutics is necessary.

## 6. Conclusions

The role of cellular senescence in skin pigmentation during the aging process cannot be described as a single process or pathway excepting some congenital and acquired disorders. Complicated intercellular interactions may be involved in skin-aging pigmentation. Senescent cells alone or their effects on melanocytes as pigmentary cells can induce skin pigmentation in relation to aging. In this review, we suggest that the most important senescent cells in skin-aging pigmentation are fibroblasts and melanocytes and that each cell type is mainly engaged in hyperpigmentation and hypopigmentation, respectively, in aged skin. Additionally, senescent melanocytes may contribute to the senescence of neighboring skin cells. Furthermore, finding possible therapeutic targets for senescence-related materials is also important to treat and prevent skin aging and skin-aging pigmentation.

## Figures and Tables

**Figure 1 cells-11-02541-f001:**
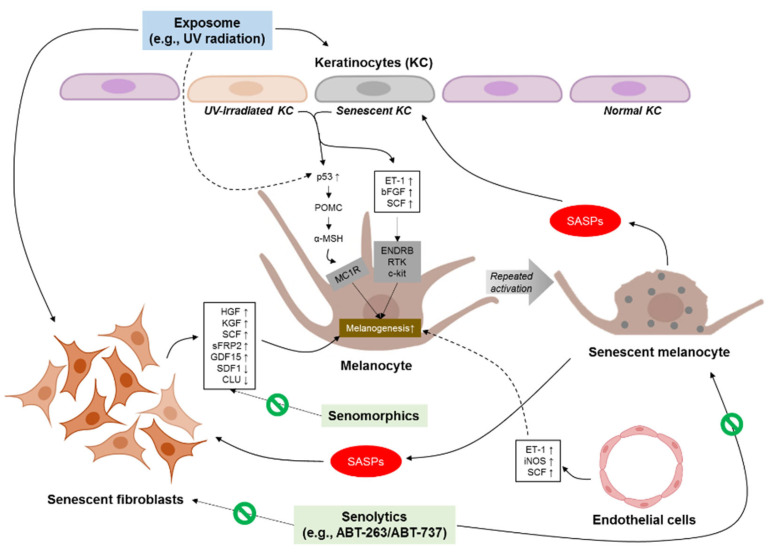
Cellular crosstalk between melanocytes and neighboring cells in skin-aging pigmentation. UV, ultraviolet; SASP, senescence-associated secretory phenotype, POMC, p53-mediated upregulation of proopiomelanocortin; α-MSH, α-melanocyte-stimulating hormone; MC1R, α-MSH stimulates the melanocortin 1 receptors; ET-1, endothelin-1; bFGF, basic fibroblast growth factor; SCF, stem cell factor; ENDRB, endothelin B receptor; RTK, receptor tyrosinase kinase; HGF, hepatocyte growth factor; KGF, keratinocyte growth factor; sFRP2, secreted frizzled-related protein 2; GDF15, growth differentiation factor 15; SDF1, stromal-derived factor 1; CLU, clusterin; iNOS, inducible nitric oxide synthase.

## Data Availability

Not applicable.

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
