# Peer review of "Skin-Aging Pigmentation: Who Is the Real Enemy?"

_cells, 2022, doi:10.3390/cells11162541_

Round 1

Reviewer 1 Report

When discussing the effects of ABT-263, the mechanism of action should be discussed.

Also, ABT-263 has been shown to decrease trabecular bone volume fraction and therefore could be potentially harmful in the treatment of age related bone changes. This should be discussed in your article.

Author Response

Point 1: When discussing the effects of ABT-263, the mechanism of action should be discussed. Also, ABT-263 has been shown to decrease trabecular bone volume fraction and therefore could be potentially harmful in the treatment of age related bone changes. This should be discussed in your article.

Response 1: Thank you for comments. According to your commnets, we added the mechanism of action and the side effect of ABT263 in revised manuscript (lines 356 and lines 364-366 in page 7-8).

Reviewer 2 Report

The current review manuscript provides a concise overview of skin-aging pigmentation. It is interesting because skin aging-related pigmentation is a commonly found skin issue. Moreover, treatment options for skin-aging pigmentation remain limited. However, the current manuscript needs to be substantially improved before being accepted for publication in Cells:

  1. The current manuscript lacks novelty compared to the recently published article by Kang et al.(Pigment Cell Melanoma Res. 2021 Jul;34(4):800-813). The authors should address the most recent findings or the progress of the relevant studies. These include current results from the clinical or investigative trials if there are any.
  2. In Conclusion (line 347), the authors concluded that the role of cellular senescence in skin pigmentation during skin pigmentation could not be described as a single process or pathway. However, a single process or pathway could be crucial for developing skin-aging pigmentation in certain congenital or acquired diseases. The authors should consider including and discussing these pathogenic skin-aging conditions. 
  3. Skin-aging-related hypopigmentation and hyperpigmentation are probably due to distinct pathways or processes, although both could coexist during skin aging. The authors should consider differentiating them in the description, as the mechanisms and therapies could differ. For example, defect in autophagy due to TSC or EPG5 mutations leads to hypopigmentation but not hyperpigmentation.
  4. Other key factors that are highly correlated with skin aging, such as sex steroid hormones and receptors, could be included in the manuscript, in addition to the cytokines as discussed in the manuscript (e.g., IL-6, IL-8, and CXCL, Line 1-5-108). 

Author Response

The current review manuscript provides a concise overview of skin-aging pigmentation. It is interesting because skin aging-related pigmentation is a commonly found skin issue. Moreover, treatment options for skin-aging pigmentation remain limited. However, the current manuscript needs to be substantially improved before being accepted for publication in Cells:

Point 1: The current manuscript lacks novelty compared to the recently published article by Kang et al.(Pigment Cell Melanoma Res. 2021 Jul;34(4):800-813). The authors should address the most recent findings or the progress of the relevant studies. These include current results from the clinical or investigative trials if there are any.

Response 1: Thank you for your comments. The review article published in PCMR has focused on the general pigmentary changes including vitiligo during the aging process. Although it was briefly mentioned in the previous paper, the current manuscript focused and addressed the cellular senescent changes in depth in aging skin. And according to your suggestion, we included recent fingding from the clinical trial in relation to therapeutic strategy for skin-aging (lines 351-354, page 7).

Point 2: In Conclusion (line 347), the authors concluded that the role of cellular senescence in skin pigmentation during skin pigmentation could not be described as a single process or pathway. However, a single process or pathway could be crucial for developing skin-aging pigmentation in certain congenital or acquired diseases. The authors should consider including and discussing these pathogenic skin-aging conditions.

Response 2: Thank you for valuable comments. According to your comments, we included those pathogenic skin-aging conditions in revised manuscript (lines 198-203, pages 4) and changed the sentences in conclusion section (line 380-382, page 8)

Point 3: Skin-aging-related hypopigmentation and hyperpigmentation are probably due to distinct pathways or processes, although both could coexist during skin aging. The authors should consider differentiating them in the description, as the mechanisms and therapies could differ. For example, defect in autophagy due to TSC or EPG5 mutations leads to hypopigmentation but not hyperpigmentation.

Response 3: Thank you for comments. It is good point. However, we considered cellular senescence and cellular crosstalk between skin cells in skin-aging pigmentation as the main subjects in this review, thus it was decribed in a way focusing on the role of each cell in skin-aging pigmentation. For this reason, we did not describe hypopigmenation and hyperpigmentation seperatly. However, according your suggestion, we added the autophagy impairment related hypopigmentation in revised manuscript (lines 237-244, page 5), because we only included autophagy impariment related hyperpigmentation in original manuscript.

Point 4: Other key factors that are highly correlated with skin aging, such as sex steroid hormones and receptors, could be included in the manuscript, in addition to the cytokines as discussed in the manuscript (e.g., IL-6, IL-8, and CXCL, Line 1-5-108).

Response 4: Thank you for comments. Accroding to your comments, we included the effect of hormonal factors on the skin aging and skin aging pigmentation such as melasma (lines 119-124, page 3).

Reviewer 3 Report

I have read with interest the review by Kim and colleagues on skin-aging pigmentation (and the hyperpigmentation related to the skin cells senescence). I believe that the review is well-written, fluent and exhaustive. The role of different cells (keratinocytes, fibroblasts, endothelial cells and melanocytes) has been properly investigated. The paper is tidy in its presentation and I do not have particular comments/criticisms.

Anyway, I only have to remark that, at the beginning of the reading, I expected the Fifth session (Therapeutic strategy for skin-aging pigmentation) to be longer and more detailed. Indeed, I believe that this is the main and more important chapter of the review. If is possible, I would appreciate to discuss more in depth the therapeutic strategies, also with some images depicting the mechanism of action. 

Morever, just a couple of tiny comments:

- line121: the verb "regress" maybe is not adequate. Did you mean that the melasma can worsen due to the estrogen underlying UVB exposure? Please clarify

- line 127: the cited article by Krutman et al also reports stress and sleep loss as external factors composing the exposome.

Author Response

I have read with interest the review by Kim and colleagues on skin-aging pigmentation (and the hyperpigmentation related to the skin cells senescence). I believe that the review is well-written, fluent and exhaustive. The role of different cells (keratinocytes, fibroblasts, endothelial cells and melanocytes) has been properly investigated. The paper is tidy in its presentation and I do not have particular comments/criticisms.

Point 1: Anyway, I only have to remark that, at the beginning of the reading, I expected the Fifth session (Therapeutic strategy for skin-aging pigmentation) to be longer and more detailed. Indeed, I believe that this is the main and more important chapter of the review. If is possible, I would appreciate to discuss more in depth the therapeutic strategies, also with some images depicting the mechanism of action.

Response 1: Thank you for comments. According to your comments, we more disccused in chaper 5 and added a figure in revised manuscript.

Point 2: Morever, just a couple of tiny comments:

- line121: the verb "regress" maybe is not adequate. Did you mean that the melasma can worsen due to the estrogen underlying UVB exposure? Please clarify

Response 2: Thank you for comments. We meaned that the melasma can regress over time because estrogen levels decrase over time. We clearly modified the sentence in the revised manuscript. (lines 119-124, page 3).

Point 3: - line 127: the cited article by Krutman et al also reports stress and sleep loss as external factors composing the exposome.

Response 3: Thank you for comments. According to your comments, we added the extrernal factors in revised manuscript (line 128, page 3).

Round 2

Reviewer 1 Report

Thank you for adding more information on ABT-263.

Please rewrite these sentences to make them clearer: 1). “In aspect of skin pigmentation, these hormonal changes can improve melasma, an acquired chronic relapsing hyperpigmented disease, can regress because the estrogen underlying UVB exposure can maintain pigmentation by increasing the number of blood vessels [50].” 2). “However, the causes of skin-aging pigmentation are not only restricted to melanocytes and at the cellular level, as typical  neighboring skin cells including keratinocytes, fibroblasts, and endothelial cells also interact with pigmentary cells during the disruption of the homeostasis of the pigmentation system.”

Author Response

Point 1: Thank you for adding more information on ABT-263.

Please rewrite these sentences to make them clearer: 1). “In aspect of skin pigmentation, these hormonal changes can improve melasma, an acquired chronic relapsing hyperpigmented disease, can regress because the estrogen underlying UVB exposure can maintain pigmentation by increasing the number of blood vessels [50].” 2). “However, the causes of skin-aging pigmentation are not only restricted to melanocytes and at the cellular level, as typical neighboring skin cells including keratinocytes, fibroblasts, and endothelial cells also interact with pigmentary cells during the disruption of the homeostasis of the pigmentation system.”

Response 1: Thank you for your comments. According to the suggestions, we have changeed the sentences like below.

(1) “In aspect of skin pigmentation, an acquired chronic relapsing hyperpigmented disease such as melasma can be improved as the estrogen level decreases, because the estrogen underlying UVB exposure can maintain pigmentation by increasing the number of blood vessels” (lines 121-124, page 3)

(2) “However, the causes of skin-aging pigmentation are not only restricted to melanocytes itself. Typical neighboring skin cells including keratinocytes, fibroblasts and endothelial cells also interact with pigmentary cells during the disruption of the pigmentation system’s homeostasis” (lines 195-198, page 4) 

Reviewer 2 Report

The authors have addressed the questions raised in the previous review section. The revised version of the manuscript has been dramatically improved. I would suggest correcting the minor issues in the English language, such as …the simultaneously use… (line 68). 

Author Response

Point 1: I would suggest correcting the minor issues in the English language, such as …the simultaneously use… (line 68).

Response 1: Thank you for comments. According to your suggestion, we have changed the word “simultananeously use” to “combined use.” (line 68, page 2)